# Maternal Intake of Either Fructose or the Artificial Sweetener Acesulfame-K Results in Differential and Sex-Specific Alterations in Markers of Skin Inflammation and Wound Healing Responsiveness in Mouse Offspring: A Pilot Study

**DOI:** 10.3390/nu15112534

**Published:** 2023-05-29

**Authors:** Pania E. Bridge-Comer, Mark H. Vickers, Sandra Ferraro, Aurélie Pagnon, Clare M. Reynolds, Dominique Sigaudo-Roussel

**Affiliations:** 1Liggins Institute, University of Auckland, Auckland 1023, New Zealand; pab4011@med.cornell.edu (P.E.B.-C.); m.vickers@auckland.ac.nz (M.H.V.); 2Laboratory of Tissue Biology and Therapeutic Engineering (LBTI), UMR 5305 National Center for Scientific Research (CNRS), 7 Passage du Vercors, CEDEX 7, 69367 Lyon, France; sandra.ferraro@ibcp.fr; 3UFR Biosciences, University of Claude Bernard Lyon 1, 69622 Villeurbanne CEDEX, France; 4NOVOTEC, ZAC du Chene_Europarc, 11 Rue Edison, 69500 Bron, France; apagnon@novotec-labs.com; 5School of Public Health, Physiotherapy and Sports Science, Conway Institute, Institute of Food and Health, University College Dublin, 4 Dublin, Ireland

**Keywords:** developmental programming, DOHaD, animal model, maternal nutrition, fructose, artificial sweetener, inflammation, wound healing, skin, adipose tissue, pressure injury

## Abstract

Growing evidence has demonstrated that maternal artificial sweetener (AS) consumption may not be a beneficial alternative when compared to sugar-sweetened beverages and potentially leads to metabolic dysfunction in adult offspring. Compromised skin integrity and wound healing associated with type 2 diabetes can lead to complications such as diabetic pressure injury (PI). In this context, the skin plays an important role in the maintenance of metabolic homeostasis, yet there is limited information on the influence of sugar- or AS-sweetened beverages during pregnancy on developmental programming and offspring skin homeostasis. This study examined the impact of maternal fructose or acesulfame-k consumption on offspring wound healing. Female C57Bl/6 mice received a chow diet ad libitum with either water (CD), fructose (FR; 34.7 mM fructose), or AS (AS; 12.5 mM Acesulfame-K) throughout pregnancy and lactation. PIs were induced in offspring at 9 weeks of age (*n* = 6/sex/diet). PIs and healthy skin biopsies were collected for later analysis. Maternal AS intake increased skin inflammatory markers in healthy biopsies while an FR diet increased *Tgfb* expression, and both diets induced subtle changes in inflammatory markers post-wound inducement in a sex-specific manner. Furthermore, a maternal FR diet had a significant effect on pressure wound severity and early wound healing delay, while AS maternal diet had a sex-specific effect on the course of the healing process. This study demonstrates the need for a better understanding of developmental programming as a mediator of later-life skin integrity and wound responsiveness.

## 1. Introduction

Research into the Developmental Origins of Health and Disease (DOHaD) continues to highlight the importance of early life nutritional exposures on overall health and wellbeing across the life course. A primary focus has been on outcomes related to cardiometabolic disease, although it is clear that the maternal diet impacts the development of a broad range of organ systems in offspring. There is considerable experimental evidence linking a maternal high-fat diet to obesity and metabolic dysfunction in offspring. However, there remains a relative paucity of data on maternal sugar intake in the context of DOHaD despite increasing evidence that sugar-sweetened soft drinks, particularly those high in fructose, are contributing significantly to the obesity epidemic [1,2]. Experimental evidence to date has highlighted the detrimental effects of fructose consumption during pregnancy on both mother and child [3,4,5]. In an attempt to circumvent these adverse effects, the consumption and availability of artificial sweeteners (ASs) via “diet” beverages and foods have increased considerably. While several compounds have been approved, the most widely used are acesulfame-potassium (Ace-K) and aspartame, with natural non-caloric sweeteners such as stevia also becoming popular. Despite being viewed as the “healthier” option and consumed due to their low energy content [6], there is much controversy in relation to the health effects of non-caloric sweeteners with links to cancer, neurological effects, obesity, and metabolic dysfunction [7,8,9].

Over a quarter (29.5%) of pregnant women consume AS, of which over 5% do so daily [10,11]. This rate of consumption increases in women diagnosed with gestational diabetes (GDM), where just under half (45.4%) report AS intake during pregnancy, and 9.2% report daily intake [12]. Several sweeteners, including Ace-K, can cross the placenta to the fetal circulation. Work by Laforst-Lapointe et al. showed that maternal consumption of AS during pregnancy was associated with infant gut microbiota and metabolic modifications concomitant with an increased infant body mass index [11]. A recent systematic review demonstrated that maternal exposure to AS during pregnancy is associated with an increased risk of preterm birth and increased birth weight [13]. Previous work by our group has demonstrated that exposure to Ace-K and FR results in maternal glucose intolerance and reduced pregnancy duration, as well as glucose intolerance and adipocyte hypertrophy in female but not male offspring [14,15,16].

To date, most experimental models have focused on the primary insulin-responsive tissues, including the liver, muscle, and adipose. However, a common phenotypic outcome associated with insulin resistance (IR) and type 2 diabetes (T2DM) is compromised skin integrity and poor wound healing, leading to complications such as diabetic and pressure injuries (PI). In this context, an increasing body of evidence supports skin as an important peripheral (neuro)endocrine organ that is both hormonally and metabolically active, with key roles in the maintenance of metabolic homeostasis [17,18]. Alterations in dermal and epidermal skin homeostasis [19,20,21,22] can lead to a delay or failure in wound repair, as observed in diabetic patients and animal models [23,24].

However, PI occurrence remains high despite an increase in clinical studies examining factors such as PI aetiology, improved staff training, improved understanding of diabetes management and the risks for developing PIs in patients and medical devices to help prevent PIs. As such, there is still a need to understand the biochemical factors and potential mechanistic processes that could fragilize the skin.

This pilot study in mice therefore aimed to investigate the influence of a maternal diet supplemented with AS or fructose across pregnancy and lactation on wound healing in female and male adult offspring. We hypothesized that maternal AS or fructose intake would alter female and male offspring skin integrity and wound healing following induction of PI.

## 2. Materials and Methods

### 2.1. Animal Model

All of the animal procedures were approved in accordance with the New Zealand Animal Welfare Act, 1999, by the Animal Ethics Committee at the University of Auckland (Approval number 001846). C57BL/6 mice breeding pairs from the Vernon Jansen Unit at the University of Auckland were purchased and housed under standard conditions (wood shavings as bedding, 22 °C, 40–45% humidity, and a 12 h light: 12 h dark cycle). All mice were maintained on a standard chow diet (Envigo, Indianapolis, IN, USA, 2018 Teklad Global 18% Protein Rodent Diet, Appendix A) ad libitum throughout the experiment. At 10 weeks of age, age-matched female mice were housed with an unrelated age-matched C57BL/6 male for mating. Confirmation of mating using a vaginal plug was recorded as gestation day (GD) 0.5. Following confirmation of mating, dams were randomly assigned into one of three dietary groups:(a)Control (CD; standard diet and drinking water);(b)Artificial sweetener (AS; standard diet and 12.5 mM Ace-K in drinking water);(c)Fructose (FR; standard diet and 34.7 mM fructose in drinking water).

Both AS and FR diets were calculated to be equivalent to a human dose of one standard can of soda a day, with the FR dose providing approximately 20% of the total daily caloric intake. Diets were maintained across pregnancy and lactation. The date of birth was noted and recorded as postnatal day 1 (P1). At P2, the litters were weighed, sex determined through anogenital distance, and randomly reduced to eight pups per litter (4 males and 4 females) in any litter with over eight pups in order to standardise nutrition until weaning. At weaning (P21), the offspring were housed in same-sex sibling groups and body weight and food intake were measured weekly. All of the offspring were fed the standard control diet and water ad libitum from weaning until the end of the experiment (11 weeks). The primary outcomes for this cohort related to adipose tissue dysfunction, glucose tolerance and reproductive function have been published previously [14,15,16] and serve to validate the model used. For this pilot study, a sub-cohort of animals was used (*n* = 6 per sex per dietary group). These animals were randomly selected at 9 weeks of age and studied at 11 weeks of age.

### 2.2. Pressure Injury Model

The wound model used was a modified version of a clinically relevant and reproducible mouse model as established previously [25] and detailed below.

#### 2.2.1. Preparation of Skin

Two days prior (D-2) to the beginning of skin compression, mice were individually anaesthetised using isoflurane, and an electric shaver and depilatory lotion (Veet Hair Removal Cream) were used to remove a 3–4 cm section of hair from the base of the tail up the back of the mouse. The lotion and hair were removed with a gentle swab and warm water, avoiding strong skin friction. 

#### 2.2.2. Skin Compression

Two days following shaving (D0), mice were anaesthetised using isoflurane. Magnets (Magsy, 417.8 mmHg, 8 mm diameter, 1 mm thickness) were placed on the dorsal skin (Figure 1A,B) to induce two PIs with a strip of non-wounded skin between the wounds. The magnets were applied twice for two hours with a magnet-free interval of 15 min. Photographs were taken prior to magnet placement and 3, 5, 7 and 9 days after magnet removal. Skin biopsies were taken on days 5 and 9 for further analysis.

#### 2.2.3. Characterisation of Wound Area

Wounds were observed from the photos in a blinded manner and three parameters were characterised: inflamed lesion (the inner red crown from the central point of the wound), the skin barrier breakdown area (a crust on the skin), and the ischemic crown (the outermost white crown around the wound) (Figure 1C). The wounds were visualised, and the incidence (%) of each characterisation was calculated. Samples for histology were excised in a semi-circle with a layer of healthy skin included and placed in 10% neutral buffered formalin (NBF). Following sample collection, the wound was closed using stainless steel wound clips (Reflex 9 mm, Roboz Surgical Instrument Co., Gaithersburg, MD, USA). At D9, a biopsy was taken of the remaining skin sample, as previously described, and then the mice were culled by cervical dislocation while anaesthetized.

#### 2.2.4. Histology and Immunostaining

Skin specimens were fixed with neutral buffered formalin, dehydrated, embedded in paraffin, and cut in 5 μm sections. For the histological analysis, the sections were labelled with hematoxylin–eosin–safran. For the immunohistological analysis, following deparaffinization, the antigenic sites are unmasked using a pretreatment of citrate/microwave. The sections were incubated overnight at 4 °C with the primary antibody in BPS-BSA 3% buffer (see Appendix A). Following the inhibition of endogenous peroxidases using hydrogen peroxide, the sections were incubated in the secondary antibody coupled to peroxidase [Dako, EnVision mouse, ref. K4001 or EnVision rabbit, ref. K4003]. The reaction with its substrate, diaminobenzidine (DAB) [Dako, ref. K3468], reveals the antigen–antibody complexes by the appearance of a brown stain. The sections were counterstained with Mayer’s hematoxylin then mounted between slide and coverslip in aqueous medium.

#### 2.2.5. Gene Expression Analysis

At D5 and D9, skin was collected from one of the two PI on each mouse. Samples for PCR analysis were excised by taking the entire wound with no healthy skin and placed in RNAlater. A sub-cohort of mice (*n* = 3 per sex per dietary group) were used to collect healthy skin biopsies without implementation of the PI. This sample was taken from the same area of the back following the same skin preparation, as previously described.

RNA was extracted from PI using RNeasy Mini Kits (Cat. No 74104, Qiagen, Hilden, Germany) and a TissueLyser (Qiagen, Hilden, Germany) as per the manufacturers’ instructions. Following extraction, a NanoDrop spectrophotometer (NanoPhotometer N60, Implen) was used to assess RNA. A High-Capacity cDNA Reverse Transcription Kit (Life Technologies Ltd., Applied Biosystems, Paisley, UK) was used to generate cDNA, as per the manufacturers’ instructions. PCR was performed using the Applied Biosystems QuantStudio 6 Flex Real-Time PCR system (Applied Biosystems). A panel of reference genes were assessed, and those that did not change in relation to maternal diet or offspring sex were used for normalisation. Genes (Appendix B
Table A1) were normalized to the geomean of *Rps29* and *Rps13* expression. Data were analysed using the comparative CT method [26].

#### 2.2.6. Statistical Analysis

Statistical analysis was performed with GraphPad Prism and SigmaPlot (v14). Shapiro-Wilk test was used to assess normality. Any data that were not normally distributed were transformed as appropriate. Wound characterisation was analysed using Chi-square tests. D5 and D9 PCR data were analysed using two-way ANOVA with maternal diet and day as factors. Healthy D0 PCR data were analysed using one-way ANOVA. Holm-Sidak post hoc tests were performed as indicated for testing comparisons between the groups. Significance between the groups was given at *p* < 0.05. Male and female mice were analysed separately. All of the data are presented as mean ± SEM unless otherwise stated.

## 3. Results

### 3.1. Physiological Data

Weights at birth, weaning, and cull, and plasma glucose at cull, and HOMA were recorded and reported in a previous study by our group (Appendix B
Table A2) [14].

### 3.2. Skin Characterisation Prior to Wound Induction

#### 3.2.1. Healthy Skin Gene Expression

In female offspring at D0, there was a significant increase in inflammatory gene expression for *Tnfa* in AS compared to CD and FR (Figure 2A). There was no significant effect of a maternal AS diet on the expression of *Il1b*, *Nlrp3*, *Vegfa*, *Tgfb* or *Pparg*. (Figure 2B–E,G). *Ppargc1* gene expression was significantly decreased in the FR group compared to AS (Figure 2F).

In male offspring at D0, *Tnfa, Il1b*, *Nlrp3*, *Vegfa*, *Pparg* and *Pparcg1* gene expression was not significantly altered (Figure 2A–C,F,G). AS diet in male offspring led to a significant decrease in *Tgfb* gene expression compared to CD while the expression in the FR group was significantly increased compared to AS (Figure 2E). There were no significant differences in FR males in the expression of *Vegfa*, *Ppargc1* or *Pparg* (Figure 2D,F,G) compared to CD. 

#### 3.2.2. Wound Characterisation

At D3, there was no change in pressure wound occurrence between the groups (Figure 3A). Overall, at D3, the total wound surface area was significantly larger in maternal FR diet male and female offspring compared to the CD and AS diets (Figure 3C). There was no difference in the total size of the wound between maternal AS diet and the controls. 

At D3 and D5 after magnet application, there was no difference in wound characterisation between maternal diets in female groups (Figure 4). At D3 after magnet application, there was no difference in wound characterisation for inflammation and skin breakdown between maternal diets in the male groups. 

In contrast to female wound characterisation, male offspring exhibited an increased in skin whitening around the wound in the FR diet compared to CD (Figure 4). Skin breakdown and inflammation were not significantly different between the groups.

#### 3.2.3. Wound Healing

For each maternal diet group, the size of the wounds at D were not different between males and females, and the wounds were closed at D9 with no difference between males and females (Figure 5). With regard to the temporal course of the skin wound healing process, a significant delay in wound healing was observed for the CD diet male offspring at D5. This delay between males and females was not observed in the AS and FR diet groups (Figure 5). 

Maternal AS diet did not affect the wound process compared to the CD diet (Figure 6). In male offspring, the maternal FR diet delayed the healing process to D5 without delaying the wound closing day. The same pattern is seen in the respective female offspring but did not reach statistical significance (Figure 6). Overall wound immunostaining at D5 for inflammatory markers in the maternal FR diet in female offspring revealed an increase in the semi-quantitative scoring for macrophages (CD68) in the dermis (Appendix A), which is coherent with the persistent inflammation observed in the wound characterisation (Figure 5). There was no significant difference in inflammatory wound immunostaining between diets in male offspring (Appendix A). Maternal AS diet reduced CD68 staining compared to an FR diet in both males and females at D5.

#### 3.2.4. Pressure Injury Gene Expression

*Tnfa* expression was reduced from D5 to D9 in all dietary groups in both male and female offspring. *Tnfa* expression was also reduced at day 9 in the AS and FR groups compared to CD in both female and male offspring, and there was no significant difference at day 5 between groups (Figure 7A). 

In female FR offspring, *Il1b* expression at D9 was reduced compared to D5. In male offspring, *Il1b* gene expression was significantly decreased in AS and FR compared to CD at D5 while expression in the CD group was significantly reduced at D9 compared to D5 (Figure 7B). 

There was an overall effect of the timepoint on *Nlrp3* gene expression in male and female offspring. In female offspring, *Nlrp3* expression was significantly reduced at D9 compared to D5 in CD and FR groups but not in the AS group. *Nlrp3* expression significantly decreased in CD, AS and FR groups at D5 compared to D9 (Figure 7C). 

The gene expression of *Vegfa* was reduced overall at D9 compared to D5 in both male and female offspring. In female offspring, AS and FR at D9 were also significantly reduced compared to D5 (Figure 7D).

*Tgfb* gene expression was reduced overall in both female and male offspring at D9 compared to D5 with male AS offspring having a significant reduction in expression at D9 compared to D5 (Figure 7E).

In female offspring, there was an overall dietary effect on *Ppargc1* expression but no overall difference between D5 and D9. This was reflected in a Day × Diet group interaction (Figure 7F). 

*Ppargc1* expression in females was significantly increased in the FR group at D9 compared to FR at D5 and was significantly increased at D9 in the FR group compared to the CD and AS groups. This was reflected in a Day × Diet group interaction (Figure 7F). In male offspring, there was an overall effect on diet group and timepoint on *Ppargc1* expression. *Ppargc1* gene expression was reduced between D5 and D9 overall. In the AS group, *Ppargc1* expression was reduced at D9 compared to D5. At D9, *Ppargc1* expression was significantly increased in the FR group compared to the CD and AS groups (Figure 7F). In female offspring, the gene expression of *Pparg* in PI was significantly reduced in the AS group at D5 compared to CD (Figure 7G). At D9, *Pparg* expression was significantly reduced in the CD group compared to D5 but unchanged in the AS and FR groups, as reflected in a significant overall statistical interaction. There was no overall difference in *Pparg* expression in males either within or between timepoints although there was a trend toward a reduced expression in all groups between D5 and D9 (*p* = 0.06).

## 4. Discussion

While the impact of maternal diet has been comprehensively assessed in relation to cardiometabolic health, there is little information on how developmental processes impact the skin of offspring. This pilot study, therefore, set out to examine the effect of maternal sweetener (caloric, FR and non-caloric, Ace-K) intake on offspring skin wound occurrence and severity and examine the impact of maternal diet on the quality and the integrity of skin in male and female offspring. 

Maternal Ace-K or FR intake was demonstrated to increase skin inflammatory markers in a differential manner to that of the CD group prior to wound inducement. A maternal Ace-K diet increased pro-inflammatory gene expression in offspring while an FR diet increased *Tgfb* gene expression related to anti-inflammatory processes. As a result of skin compression, an FR diet significantly increased the size of the wound with a delay in the wound healing process without delaying the closure day that was associated with subtle changes in gene expression of inflammatory markers but with significant inflammatory changes within the tissue related to macrophage infiltration. In contrast, for the maternal Ace-K offspring, the size of the wound was not different from the control group, with a decrease in inflammatory markers.

In the literature, both the maternal intake of FR and AS have been associated with a range of adverse outcomes in offspring, including glucose intolerance, T2DM and obesity [3,27,28]. Given the increasing recognition of skin as a hormonally and metabolically active organ and the well-established linkages between metabolic homeostasis, skin integrity and wound healing, the present study sought to investigate whether there was a programming component to outcomes related to wound healing in offspring. 

Inflammation plays an essential role in wound occurrence and healing processes, preventing irritants and pathogens from infecting the wound while the skin barrier is impaired. The length and intensity of the inflammatory stage, however, can impact progression to the next phase of wound healing. For example, chronic injuries are characterised by increased concentrations of neutrophils that impair extracellular matrix restructuring and contribute to the continuousness of the wound [29]. Dermal adipocytes, located within the hypodermis, play several roles in wound repair, including communication with fibroblasts and the inducement of inflammation through recruitment of inflammatory factors and key immune cells such as macrophages [30]. In a previous study by our group using this model, maternal Ace-K and FR-intake increased adipocyte hypertrophy in gonadal adipose tissue, while subcutaneous adipose tissue was found to be hypertrophic in the female offspring of Ace-K-fed mothers [14]. Adipocyte hypertrophy is a common predictor of adipose metabolic disease and is associated with the increased recruitment of pro-inflammatory adipokines [31]. This aligns with a general pattern of increased inflammatory parameters as seen in the present study with the increase in *Tnfa* in the healthy skin of Ace-K female offspring that could indicate increased epidermal inflammation at D0. Indeed, Tnfα induces an inflammatory effect within the skin [32], and has been found to be overexpressed in conditions such as psoriasis, where it has been suggested to inhibit the expression of proteins involved in the skin barrier [33]. There were differential effects observed in female offspring as compared to males as they expressed lower or no change in inflammatory gene expression compared to the control group. 

In male but not female mice at D0, *Nlrp3* expression was reduced in FR compared to CD, *Nlrp3* is expressed in human keratinocytes. In addition, *Tgfβ* gene expression was increased in FR male offspring and it is thought to be anti-inflammatory but is also essential for the homeostasis of the epithelial skin layer and has increased expression in response to acute wounds in the epidermis [34]. Thus, prior to wound induction, changes in the skin phenotype were expressed differently depending on the maternal diet and the sex of the offspring. This could influence wound occurrence, severity and healing. Surprisingly, maternal AS diet increased proinflammatory gene expression in female offspring compared to maternal FR diet. In addition, maternal Ace-K consumption increased *Pparcg1* expression compared to FR. *Ppargc1* encodes PGC-1α, which plays a key role in mitochondrial biogenesis regulation and metabolism and energy homeostasis [35], which could influence the wound healing process.

The characterisation of the wounds revealed that maternal dietary exposure had no effect on PI occurrence but there were differential effects in relation to the size of the skin lesion with FR diet inducing larger pressure wounds. Inflammation is present at the wound area in male and female offspring but only males exhibited skin whitening following wound inducement, indicating sex-specific responses to pressure injury based on the maternal dietary background that may indicate involvement of processes related to fibrosis or ischemia. The skin barrier is the uppermost layer of the epidermis and prevents loss of water and invasion by foreign irritants. Genetic and environmental factors contribute to barrier breakdown in non-wounded individuals, which can in turn induce the production of inflammatory mediators that exacerbate the cycle [36]. Wound healing follows four overlapping phases; hemostasis, marked by vascular constriction and clot formation; inflammatory, where neutrophils, macrophages and other inflammatory cells infiltrate the wound, clearing away invading microbes and apoptotic cells, and stimulating others to promote tissue regeneration as wound healing shifts towards the next stage; proliferative, characterised by re-epithelialisation, and remodelling; the final stage where remodelling of the extracellular matrix may take years [37]. Normal wound healing follows these steps in a timely manner, however chronic wounds are characterised by impaired wound healing where skin breakdown leads to further delays in healing and creates a cycle of reoccurring breakdown and healing. 

Neither Ace-K nor FR exposure in utero resulted in wound closure delay but it induced a different pattern of healing process at D5 compared to the control diet. The FR diet resulted in a significant delay on D5 for both sexes compared to their respective control group. This result is in part concordant with previous studies focusing on the effect of high glucose levels and related to diabetes [24,38] with the severity of the size of the pressure wound and an early wound healing delay but it did not recapitulate the severe wound healing delay for closure reported in these diabetic studies.

Skin gene expression was analyzed during the healing process, and most markers were significantly different between the two time points D3 and D5, where at D9, markers associated with inflammation: *Il1b, Nlrp3, Tnfa*, and *Vegfa*, gene expression, were reduced compared to D5, potentially indicating the natural progression from an increased inflammatory stage at D5 to the proliferative phase of wound healing. The expression of *Il1β* was also downregulated through maternal Ace-K and FR consumption at D5 in comparison to CD in male offspring. IL-1β is a key pro-inflammatory marker within the skin, which is often upregulated in chronic inflammatory conditions such as psoriasis, and can work in synergy with other inflammatory markers, such as Tnfα, to amplify the response [39]. Expressed by keratinocytes and various immune cells, Vegfa is also essential in mediating the processes involved in wound healing such as angiogenesis and inflammation [40]. It is upregulated in response to injuries to the skin, with protein reported to peak around day 3–5, agreeing with the gene expression data reported in this study. Conversely, abnormally low expression has been reported in individuals suffering chronic wounds, such as in diabetics [41]. Pparγ is also involved in inflammatory responses, however as an inhibitor of inflammation, and regulates cell proliferation, differentiation, and skin barrier homeostasis [42]. However, in mice, deficiencies in Pparγ have been shown to have little influence on wound healing [43], indicating that the reduction in gene expression in the D5 AS and D9 CD female offspring compared to D5 CD may not be a large contributor to the wound healing process. Collectively, the data presented here display only subtle changes. Yet, it is clear nonetheless that these alterations to maternal nutrition do program offspring skin integrity in different manners and influence wound healing. It would be worthwhile investigating more molecular pathways involved in skin wound healing and chronic injuries to further shed light on this novel area of DOHaD. 

There are some limitations to acknowledge in this pilot study. Firstly, this study was limited in terms of the number of mice per group. Increasing the experimental power of the study would help to further confirm the results seen. In addition, it would be of interest to collect skin samples from offspring at an older age to determine whether natural skin aging and fragility is impacted by the maternal dietary effect. The effects of nutritional insults that occur during gestation or early life can often have subtle effects on the health of offspring; however, these altered developmental environments may prime the offspring for later dysfunction. It is possible that by assessing these mice at an older age and/or by inducing a secondary insult, such as a high-fat diet, any potential dysfunction observed may be exacerbated. In the present study, the offspring were fed a standard control diet post-weaning and it would be interesting to examine the offspring in the setting of a postnatal obesogenic diet, which is known to amplify the programmed phenotype [44]. This pilot study also focused on the genes associated with inflammation within the skin in line with previous work undertaken by this group assessing adipose tissue. However, it would be beneficial to also investigate other molecular pathways associated with skin barrier breakdown such as filaggrin, protease and protease inhibitors. These genes are commonly altered in cases of atopic eczema and lead to increased skin barrier breakdown [36], with high protease concentrations characteristic of chronic wounds. Finally, the undertaking of further histological and immunohistochemical analysis will extend the findings reported in this pilot study. Nonetheless, the present study does provide novel data linking maternal dietary exposures with altered gene expression and wound healing responsiveness in the skin of offspring. 

In summary, Ace-K and fructose appeared to differentially alter the expression of inflammatory skin markers in offspring that could affect skin integrity and post-wound inducement. Further sex-specific changes indicate different developmental programming responses to the maternal diets in male and female offspring. 

To our knowledge, this is the first report of maternal nutrition impacting skin inflammatory responses and differential wound healing responsiveness in offspring. The present study also adds to the evidence to date, suggesting that Ace-K consumption over pregnancy and lactation did not reproduce the effect of FR consumption but itself caused differential effects as compared to the control group. Further work in the setting of developmental programming as an early mediator of later skin health and function is warranted.

## Figures and Tables

**Figure 1 nutrients-15-02534-f001:**
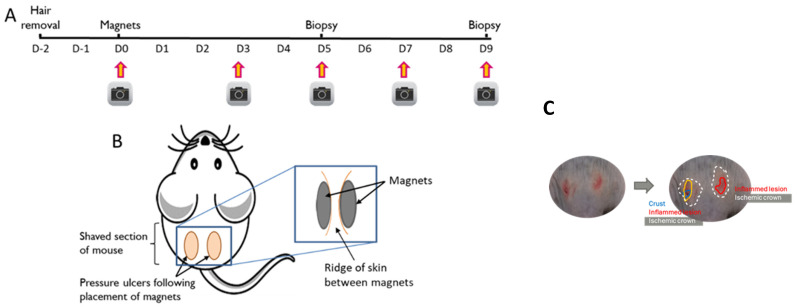
Pressure injury timeline and placement. (**A**) timeline of magnet placement, (**B**) Placement of magnets on mouse to induce pressure injury, (**C**) Method to characterise skin injury in male and female mice offspring. Protocol modified from Stadler et al. [25].

**Figure 2 nutrients-15-02534-f002:**
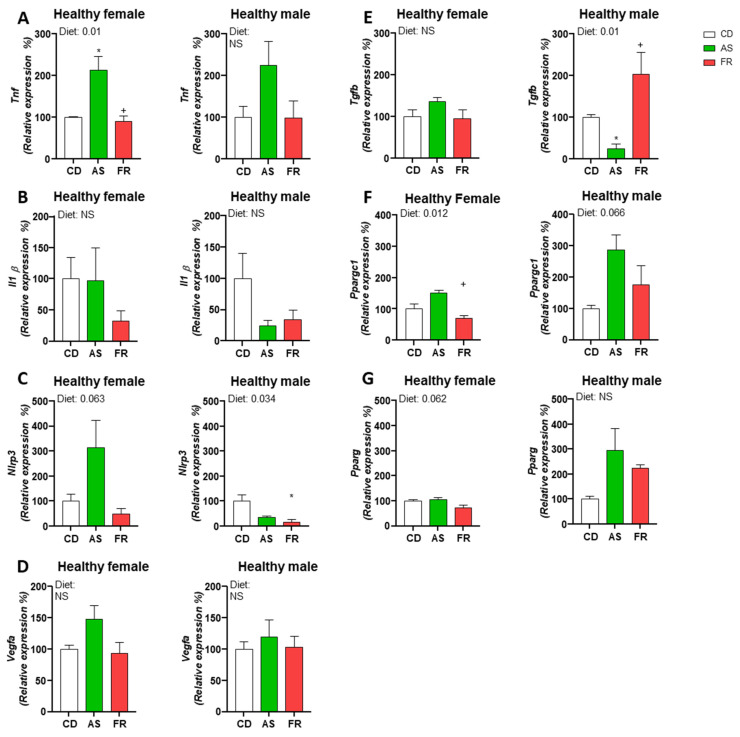
Gene expression in healthy skin. The impact of Ace-K (AS) and fructose (FR) intake compared to controls (CD) at D0 on skin gene expression in female and male offspring. (**A**) *Tnfα*, (**B**) *IL1b*, (**C**) *Nlrp3*, (**D**) *Vegfa*, (**E**) *Tgfb*, (**F**) *Ppargc1*, (**G**) *Pparg*. Data were analysed using one-way ANOVA. Data are expressed as mean ± SEM. * *p* < 0.05 w.r.t CD. + *p* < 0.05 w.r.t AS. *n* = 3/group. NS: Not significant.

**Figure 3 nutrients-15-02534-f003:**
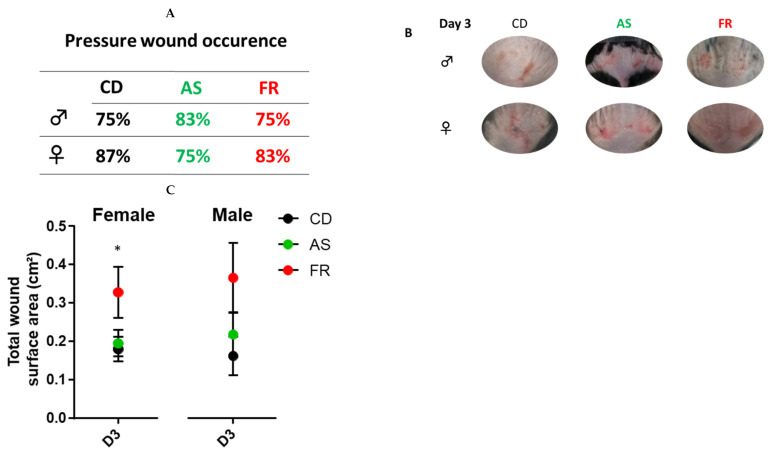
Wound characterisation at D3. The impact of Ace-K (AS) or fructose (FR) intake compared to Controls (CD) at D3 on wound characterisation in female and male C57BL/6 offspring. (**A**) Percentage of pressure injury occurrence (**B**) Representative images of wounds taken at D3 in male offspring (top) and female offspring (below). (**C**) Total surface area of pressure injury in female and male offspring; *n* = 12 wounds per group (2 wounds per mouse). * *p* < 0.05 vs. CD and AS.

**Figure 4 nutrients-15-02534-f004:**
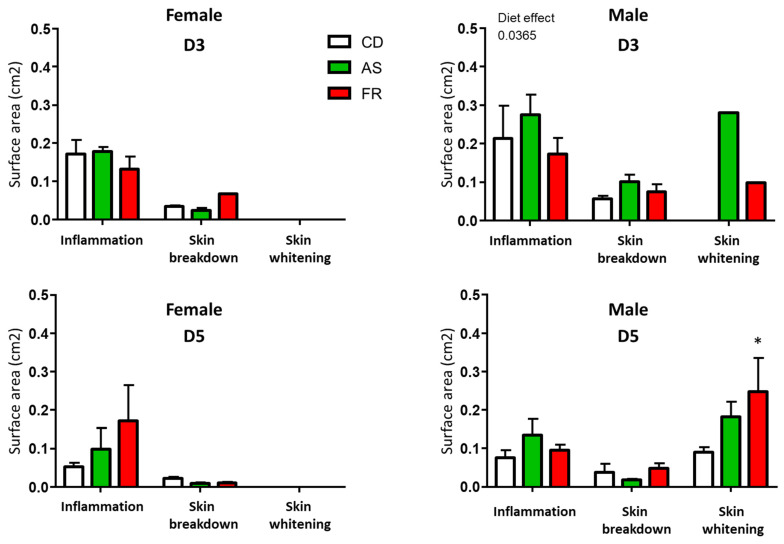
Characterisation of surface area of inflamed lesions, skin breakdown, or ischemic crowns in female and male offspring at D3 and D5. * *p* < 0.05 vs. CD.

**Figure 5 nutrients-15-02534-f005:**
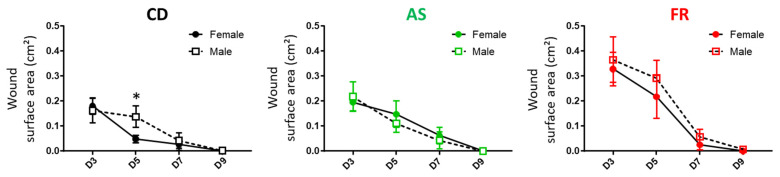
Sex-specific effects of the maternal diet on the wound healing process in offspring. * *p* < 0.05 vs. female.

**Figure 6 nutrients-15-02534-f006:**
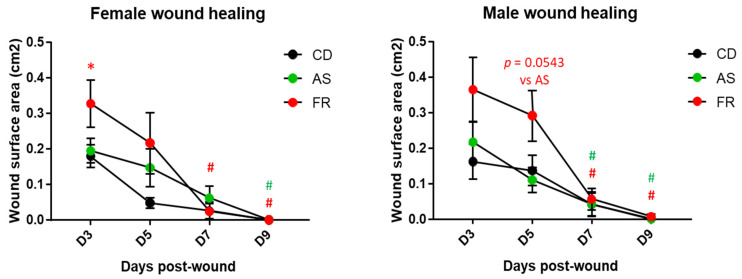
Maternal diet effect on the wound healing process in female and male offspring. * *p* < 0.05 vs. CD and vs. AS; # *p* < 0.05 vs. respective D3, red FR and green AS.

**Figure 7 nutrients-15-02534-f007:**
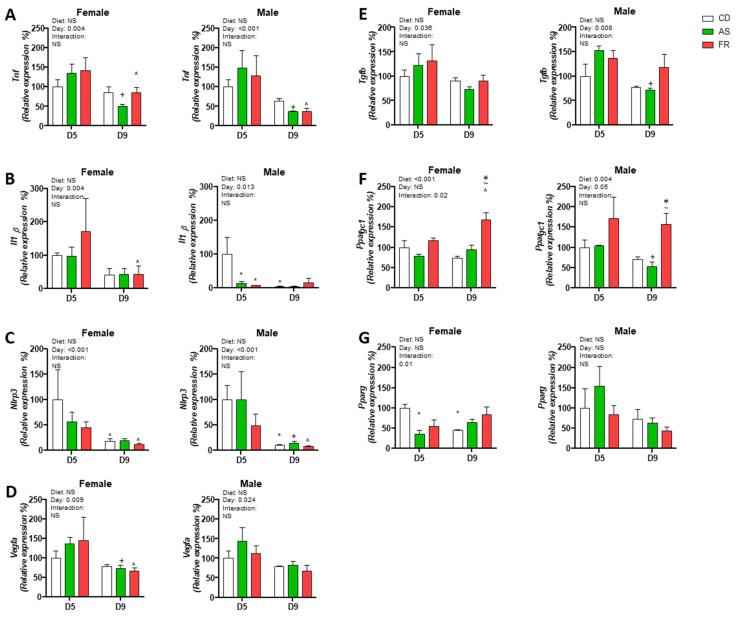
D5 and D9 pressure injury gene expression. The impact of Ace-K (AS) and fructose (FR) intake compared to D5 Controls (CD) at D5 and D9 on skin pressure injury gene expression in female and male C57BL/6 offspring mice. (**A**) *Tnfα*, (**B**) *I11b*, (**C**) *Nlrp3*, (**D**) *Vegfa*, (**E**) *Tgfb*, (**F**) *Ppargc1*, (**G**) *Pparg*. Data were analysed using two-way ANOVA. Data are expressed as mean ± SEM. * *p* < 0.05 w.r.t CD D5. + *p* < 0.05 w.r.t AS D5. ^ *p* < 0.05 w.r.t FR D5. # *p* < 0.05 w.r.t CD D9. ~ *p* < 0.05 w.r.t AS D9. *n* = 4–5/group. NS = Not significant.

## Data Availability

Data are available from the authors upon request.

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
