# Peer review of "Maternal Intake of Either Fructose or the Artificial Sweetener Acesulfame-K Results in Differential and Sex-Specific Alterations in Markers of Skin Inflammation and Wound Healing Responsiveness in Mouse Offspring: A Pilot Study"

_nutrients, 2023, doi:10.3390/nu15112534_

Round 1

Reviewer 1 Report

The manuscript authored by Bridge-Comer and colleagues, titled "Maternal intake of either fructose or the artificial sweetener acesulfame-k results in differential and sex-specific alterations in markers of skin inflammation and wound healing responsiveness in offspring: a pilot study," primarily assessed the effects of maternal fructose or acesulfame-k consumption on the wound healing process in offspring. The manuscript is well written, captivating, and easily comprehensible. The experimental design, methodology, statistical analysis, and presentation of results are all compelling. Nonetheless, there are some minor changes that the authors could make to improve the manuscript. For instance, it would be appropriate for the authors to mention the animal species in the manuscript title. Furthermore, all gene names should be italicized to conform with scientific writing conventions.

Reviewer 2 Report

In this manuscript, Bridge-Comer and co-authors describe the result of an in vivo pilot study set out to examine the effect of maternal sweetener intake (caloric, FR and non-caloric, Ace-K) on offspring skin wound occurrence and severity and examine the impact of maternal diet on the quality and the integrity of skin in male and female offspring. These results complete the description of a very interesting set of in vivo parameters used by the same group to publish other reports [ref 14-16 of the manuscript].

This reviewer personally enjoyed LOOKING AT the results collected in this in vivo study, especially because they well represent the complexity of this scientific field, that is always characterized by dimorphic phenotypes and divergence in gene expression profiles.

Unfortunately, THE READING of this manuscript was, on the contrary, unpleasant. The comments of the authors to their results, and the overall discussion and conclusions did not reflect the complexity of the scenario analyzed. The authors force an underlining take home message, unsupported by their data,   that both caloric, FR and non-caloric, Ace-K sweetener exert negative effects on the offspring, leading the readers  to a demonized image of these products. For this reason, this reviewer believes that the manuscript requires extensive rewriting before moving on and be accepted for publication.

Most of the problems can be found in the results and discussion sections. Just as examples, in the attempts to underline the negative effects of both Fr and AS, the authors discuss their results without pointing out that for many parameters, AS avoids the occurrence of the negative alterations induced by Fr. As first, the use of AS protects from the enlargement of the wound surface promoted by Fr, in both male and female progeny (Figure 3). The same is true for CD68 macrophage infiltration in the dermis (Supplementary Figure S1). Regarding gene expression analysis, AS does not reduce Nlrp3 expression in male and female offspring.

Thus,  the results and discussion sections must be fully rewritten to avoid scientifically unsupported and misleading messages.

This reviewer found these other points throughout the text

Abstract

Line 29- The abbreviation PU is not explained. Was it PI?

Lines 29-34.These lines are already misleading because the effect of Fr and of Ace-K is not always superimposable and Ace-K is sometimes able to not alter or even revert the those altered  by Fr.

Introduction

Line 46: The broad readership of nutrients would enjoy reading examples of drinks high in fructose.

Line 67: In the attempts to underline the negative effects of AS, the authors discuss one of their previously published results. However, the reference the authors refer to, indicates that glucose intolerance and adipocyte hypertrophy occurs only in FEMALE offspring but not in MALE. This information must be better delivered, and the authors should avoid leaving hidden messages in the results section.

Line 77-80: These sentences are way too scholastic. Impaired wound healing in diabetic patients involves many more factors than inflammation of the skin (Oxidative stress- Albumin rich exudates- iron overload). The study of inherited factors predisposing to skin inflammation is important enough and does not require overstating its involvement in impaired wound healing in diabetic patients.

Line 84-86: Misleading message. The results here presented point to AS and Fr resulting most of the time in different outcomes.

Materials and methods

Line 152- There is no Supplementary Table 1- The authors must give indication on the antibodies they used including code and dilutions.

Line 172- Why Rps29 and Rps13 were used as housekeeping genes for normalization?

The Quality of English language is fine.

Round 2

Reviewer 2 Report

The authors addressed all my concerns. The manuscript can now be accepted for publication.

fixing minor English spelling typos

Author Response

The reviewer has recommended acceptance of the revised manuscript. A further proofing for English and grammar has also been completed.